# Enhancement of Thermal Management Performance of Copper Foil Using Additive–Free Graphene Coating

**DOI:** 10.3390/polym16131872

**Published:** 2024-06-30

**Authors:** Bing Hu, Huilin Yuan, Guohua Chen

**Affiliations:** College Materials Science and Engineering, Huaqiao University, 668 Jimei Blvd, Xiamen 361000, China; binguuh@163.com (B.H.); yuanhlll@163.com (H.Y.)

**Keywords:** graphene, thermal conductivity, copper foil, thermal management

## Abstract

Advanced thermal interface materials with high thermal conductivity are crucial for addressing the heat dissipation issue in high-power, highly integrated electronic devices. One great potential way in this field is to take advantage of cooling copper foil (Cu) materials based on graphene (G). However, the current manufacturing of these cooling copper foil materials is accompanied by high cost, process complexity, and environmental problems, which limit their development and application. In this work, a simple, low-cost, environmentally friendly graphene-copper foil composite film (rGO/G-Cu) with high thermal conductivity was successfully prepared using graphene oxide directly as a dispersant and binder of graphene coating. The microstructure characterization, thermal conductivity and thermal management performance tests were carried out on the composite films. The results demonstrate that compared to pure copper foil (342.47 W·m^−1^·K^−1^) and 10% PVA/G-Cu (367.98 W·m^−1^·K^−1^) with polyvinyl alcohol as a binder, 10% rGO/G-Cu exhibits better thermal conductivity (414.56 W·m^−1^·K^−1^). The introduction of two-dimensional graphene oxide effectively enhances the adhesion between the coating and the copper foil while greatly improving its thermal conductivity. Furthermore, experimental results indicate that rGO/G-Cu exhibits excellent heat transfer performance and flexibility. This work is highly relevant to the development of economical and environmentally friendly materials with high thermal conductivity to meet the increasing demand for heat dissipation.

## 1. Introduction

With the rapid development of 5G, big data, clouds, and AI, high-performance and high-energy-density electronics like 5G network base stations, big data centers, electric vehicle battery systems, CPU chips, etc., face severe challenges and hazards [1,2,3,4,5]. These include high heat flows, unequal temperature distribution, and local overheating, resulting in drastically reduced operational efficiency, safety, reliability and durability, and even fire and explosions [6,7,8,9]. Therefore, excellent thermal management systems are urgently required to ensure the long-term safe and reliable operation of electronic devices. High-performance thermal conductive materials play a crucial role in the thermal management systems of electronic devices.

Graphene, the novel, emerging two-dimensional carbon material, has a honeycomb-like structure and exhibits the highest thermal conductivity among all known materials (5300 W·m^−1^·K^−1^) [10,11,12,13,14]. As a result, it is frequently combined with copper foil, which is widely used in thermal management systems, to fabricate graphene-coated copper foil with high thermal conductivity for efficient heat dissipation [15,16,17,18,19,20,21,22]. However, owing to its substantial specific surface area and tendency to agglomerate, the uniformity of the graphene coating and its various properties are greatly influenced. The weak Van der Waals forces between the layers lead to weak interlayer interactions, making the graphene coating prone to slipping and cracking [15,23,24]. To address these challenges, conventional approaches involve incorporating one or more dispersants [25,26,27,28,29,30,31] (such as carboxymethyl cellulose, sodium dodecylbenzene sulfonate, polyvinyl alcohol, and polyethylene glycol) or adhesives (including epoxy resin and acrylic resin) into the graphene coating to ensure homogeneous dispersion of graphene within the coating and stable bonding to the copper foil [15,21]. Unfortunately, this requires the use of polymeric dispersants or adhesives, which not only markedly reduce thermal conductivity but also increase the cost and raise environmental concerns [32,33]. Moreover, the current manufacturing processes for high-thermal-conductivity graphene-coated copper foil often require specialized equipment and harsh conditions (such as high temperature or pressure), resulting in elevated expenses and energy consumption [15,34,35,36]. Given today’s imperative focus on sustainability across industries under “carbon neutral” circumstances, optimization is an urgent requirement.

As is widely acknowledged, graphene oxide, derived from graphene, has abundant hydroxyl, carboxyl, and epoxy groups on its edges and surfaces. These oxygen-containing functional groups serve as ideal sites for the functionalization of graphene oxide, enabling the fixation of various active substances through covalent or noncovalent bonds. Consequently, graphene oxide exhibits remarkable adhesion and surface activity [37,38,39,40]. Furthermore, studies have indicated that the presence of a Cu substrate facilitates the reduction of graphene oxide [41]. The aforementioned strategy presents novel prospects for material design. This paper presents a highly effective approach for the preparation of graphene-coated copper foil with high thermal conductivity (rGO/G-Cu) by utilizing graphene oxide (GO). GO is employed as both a surface-active agent and a binder. The results demonstrate that a 10% rGO/G-Cu film exhibited excellent thermal conductivity (414.56 W·m^−1^·K^−1^), along with desirable flexibility. Additionally, infrared thermal imaging confirmed that the heat transfer capacity of 10% rGO/G-Cu outperformed conventional copper foil and 10% PVA/G-Cu composite film. This research provides a promising pathway for the development of environmentally friendly, cost-effective, and efficient materials for thermal management.

## 2. Materials and Methods

### 2.1. Materials

Graphene was obtained from Xiamen Knano Graphene Technology Co., Ltd. (Xiamen, China) Anhydrous ethanol (EtOH, AR), potassium permanganate (KMnO_4_, AP), sulfuric acid (H_2_SO_4_, 98 wt.%), and hydrogen peroxide (H_2_O_2_, 30% aq.) were purchased from Sinopharm Chemical Reagent Co., Ltd. (Ningbo, China) Natural flake graphite (8000 mesh; purity > 99%) was provided by Qingdao Tianhe Graphite Co., Ltd. (Qingdao, China) Polyvinyl alcohol: PVA-1788 (polymerization degree: 1700 ± 50, hydrolysis: 88%, CP) was offered by Shanghai Aladdin Biochemical Technology Co., Ltd. (Shanghai, China)

### 2.2. Preparation of GO

Graphene oxide (GO) was prepared with a modified Hummers’ method from natural flake graphite powders [42]. Firstly, 8000-mesh flake graphite (4 g) mixed with 98 wt% H_2_SO_4_ (200 mL) was stirred in a reactor in an ice water bath. The reaction mixture was maintained at this temperature for 30 min under continuous stirring. The reactor with the mixture was then stirred at 45 °C for 2 h. Subsequently, 300 mL of deionized water was gradually decanted into the solution. During the dropwise addition of 30 mL H_2_O_2_ (aq. 30%), the solution changed to a bright yellow color. Finally, 500 mL deionized water was poured into the reactor. The sample obtained was centrifuged and washed with deionized water to achieve a natural pH value. 

### 2.3. Preparation of rGO/G-Cu Composite Films

Typically, the graphene (900 mg) was initially dispersed in anhydrous ethanol (200 mL) and exposed to high-frequency ultrasound (20 kHz) for 15 min at room temperature. GO (100 mg) was dispersed in deionized water (50 mL) by high-frequency ultrasound for 10 min at room temperature, resulting in a clear GO solution. In this process, both graphene and GO were completely exfoliated into near-individual sheets. Then, GO dispersion was dripped into the graphene dispersion slowly under vigorous agitation to enhance the interaction between graphene and graphene oxide layers. The mixed dispersion was stirred at room temperature for 24 h until homogeneous distribution was achieved. It was then uniformly coated onto a 47 µm copper foil and dried using vacuum drying at 95 °C for 4 h. Under these conditions, the GO in the coating was reduced to rGO because of the catalytic action of the copper substrate to form prefabricated composite films (pre-10% rGO/G-Cu) [41]. Finally, 10% rGO/G-Cu with a dense rGO/G coating was obtained via rolling compression. The mass fraction (wt%) of GO (rGO) was calculated by the following equation:(1)$$x\;wt\%=\frac{MGO}{MGO+MG}\times 100\%=\frac{MGO}{MrGO+MG}\times 100\%$$

Here M*GO*, M*G*, and M*rGO* represent the mass of GO, graphene (G), and rGO, respectively. Then, the mass fractions of rGO in the graphene coating are 5, 10, 20, and 50 wt%, respectively. Figure 1 shows the preparation process of rGO/G-Cu composite films. PVA/G-Cu composite film was prepared using the same method as a comparison. (Polyvinyl alcohol (PVA) needs to be pre-dissolved in 85 °C deionized water, then cooled to room temperature.)

### 2.4. Characterization

The morphology and microstructures of graphene were observed using scanning electron microscopy (SEM, ZEISS Sigma 300, purchased from Carl Zeiss AG, a German manufacturer of optical and optoelectronic equipment based in Oberkochen) and transmission electron microscopy (TEM, Thermo Scientific Talos F200X G2, 200 KV, purchased from Thermo Fisher Scientific, an American technology services provider based in Waltham, Massachusetts). X-ray diffraction (XRD) was performed with a SmartLa diffractometer with Cu Kα radiation (λ = 1.5418 Å). Atomic force microscopy (AFM) images of GO were taken in the tapping mode with a Bruker Multimode 8 apparatus. X-ray photoelectron spectroscopy (XPS) was performed using a Thermo Scientific K-Alpha. The transient “laser flash” method (Nanoflash LFA 457) was employed to measure the thermal diffusivity (α) of various samples. The thermal conductivity of various films was determined by calculating κ = α·ρ·Cp, where α represents the thermal diffusivity, Cp signifies the heat capacity, and ρ denotes the mass density. The temperature of the samples was monitored over time using an infrared thermograph (Fluke Ti450, purchased from Fluke Electronic Instrumentation Company, a high quality electronic instrumentation products supplier based in Washington State, USA).

## 3. Results and Discussion 

### 3.1. Morphology and Structure of GO and G

Prior to the integration with copper foil, GO and graphene powders were stripped by high-frequency ultrasound in water and ethanol, respectively. SEM images of GO (Figure 2a) and graphene (Figure 2d) reveal that prior to the stripping process, the GO powder exhibited a large cross-linked morphology due to dehydration of its surface-rich oxygen-containing functional groups during freeze-drying. In contrast, the graphene powder agglomerated into flaky stacks owing to its exceptionally high surface energy. As shown in Appendix A, the weak D peak represents *sp^3^* carbon in the defect domain, the sharp G peak represents *sp^2^* carbon in the graphite lattice domain, and the intensity ratio of peak D to peak G (I_D_/I_G_) is used to evaluate the defect grade of carbon materials [43]. The I_D_/I_G_ value of GO is as high as 0.85, indicating a large number of defects due to oxidation. In contrast, the graphene has an extremely low I_D_/I_G_ value of 0.06, indicating high-quality graphene with fewer defects. TEM (Figure 2b) and AFM (Figure 2c) characterizations of GO demonstrate that the flakes had a diameter of approximately 2 µm and a thickness of 1 nm, providing compelling evidence for near-monolayered GO [44,45]. TEM characterization of graphene (Figure 2e) indicated that its flakes also had a diameter of approximately 2 µm, while the resulting electron diffraction pattern rings (Figure 2f) display the typical characteristics of near-monolayered graphene (single or double layer) [46]. These findings suggest that both GO and graphene obtained after stripping possess similar near-monolayer structures, facilitating their uniform recombination.

### 3.2. Morphology and Structure of rGO/G-Cu and 10% PVA/G-Cu

The microstructure of the graphene coating and copper foil composite was analyzed. For this purpose, scanning electron microscopy (SEM) and energy-dispersive X-ray spectroscopy (EDS) were used. As shown in Figure 3 and Appendix A, the composite films with different GO contents exhibited uniformly flat and smooth surfaces, while those with polyvinyl alcohol (PVA) as the adhesive displayed a less regular surface profile and were observed to contain defects and fragments. This phenomenon can be attributed to the linear structure of PVA, which made it difficult to interact with the lamellar structure of graphene, resulting in a less pronounced bonding effect. In contrast, GO, which has a two-dimensional lamellar structure similar to graphene, forms a compatible interface that facilitates the formation of a stable composite formation [47]. Consequently, this results in an unobstructed thermal conductivity path within the material, which enhances thermal conductivity. EDS mapping shows that there was a uniform distribution of C and O on the surface of all composite films containing different amounts of GO, indicating that both GO and graphene were present homogeneously in the films. As the GO content decreased, the concentration of O on the surface of the composite material also decreased, which is consistent with the EDS analysis of the elements (Appendix A). This controlled decrease contributes to an improvement in thermal conductivity while maintaining the material stability. Conversely, for composites with 10 wt% PVA as adhesive, an uneven distribution and agglomeration of O elements were observed on the surface of PVA/G-Cu. This is attributed to potential entanglement issues associated with linear polymers, which lead to aggregation of graphene sheets, thereby reducing their thermal conductivity [48].

The 10% rGO/G-Cu composite film serves as a representative example, as shown in Figure 4a. From the cross section, it can be clearly seen that the composite film consists of a 9 μm thick graphene coating and a 40 μm thickness copper foil, and the connection between the copper foil and the graphene coating is very strong. Furthermore, the composite films exhibited a smooth silver-gray metallic luster and considerable flexibility (Appendix A). This result shows that GO plays a good “glue” role in the composite films. The phase composition of the composite films was characterized using XRD, and the results are shown in Figure 4b. The diffraction peak at 2θ = 26.1 is the characteristic crystal face peak (002) of graphene [14]. Two strong diffraction peaks appear at 2θ values of 50.1° and 73.8°, corresponding to the (200) and (220) crystal faces of face-centered cubic Cu [35]. Remarkably, the characteristic peak of GO was not observed in the XRD pattern. This absence can be attributed to the reduction of GO during the preparation of the composite films, which is facilitated by the catalytic activity of copper [41]. In addition, a small amount of Cu_2_O was found in rGO/G-Cu composite films, but this situation was not reflected in PVA/G-Cu composite film, which indicates that compared with PVA, GO as a binder will effectively promote the bonding between graphene layer and Cu matrix, providing an effective heat conduction pathway. The chemical composition and bonding characteristics of the composite films were further investigated using XPS. From the XPS survey spectra of the composite film and GO (Figure 4c), it is evident that the C/O ratio in the composite film is enormously higher than that in GO, which can be attributed to the abundant presence of graphene in the composite films, leading to a remarkable enhancement in thermal conductivity. Figure 4d illustrates the broad C 1s spectrum of GO, which was fitted into four peaks. Among them, a binding energy of 284.8 eV corresponds to C-C bonds within the GO skeleton, while three other binding energies at 286.6 eV, 287.2 eV, and 288.6 eV correspond to C-O, C=O, and O-C=O bonds present at defect sites on GO, respectively [45,49]. In comparison with GO, only C-C and C-O bonds were observed in the broad C 1s spectrum of 10% rGO/G-Cu (Figure 4e), indicating that a certain degree of reduction occurred during the preparation process for these composite films, as well as effective repair of GO defects. The results presented in Figure 4f demonstrate that PVA/G-Cu retained its initial chemical composition throughout the preparation process and exhibited structural stability, as evidenced by the presence of both PVA and graphene components, in accordance with the findings from the XRD and SEM analyses.

### 3.3. Thermal Conductivity of rGO/G-Cu and PVA/G-Cu

Figure 5a shows the thermal conductivity of rGO/G-Cu as a function of the GO mass fraction in the coating. The thermal conductivity of the rGO/G-Cu films exhibited a linear increase with decreasing GO mass fraction of GO in the coating. This suggests that the majority of introduced graphene coatings play an important role in forming efficient heat conduction paths within the rGO/G-Cu composites. Notably, a significant improvement in thermal conductivity was observed when the GO mass fraction was 10%. The reason for this is the increase in the higher graphene content with fewer defects, which may be associated with a reduction in the amount of GO present in the coating. At a GO content of 10 wt%, an optimal proportion was achieved, which promoted the interaction and uniform dispersion between GO and graphene, resulting in the formation of a continuous thermal conductivity channel and achieving excellent thermal conductivity. However, insufficient amounts of GO cannot be completely and uniformly bonded with excess graphene, resulting in reduced thermal conductivity at a GO content of 5 wt%. This finding is consistent with the results of the EDS mapping analysis (Figure 3 and Appendix A). To further investigate the thermal conductivity of the composite films, we also studied 10% PVA/G-Cu prepared using similar methods with polyvinyl alcohol (PVA) as a binder and surfactant. As shown in Figure 5b, the thermal conductivity of 10% rGO/G-Cu and 10% PVA/G-Cu are 414.56 W·m^−1^·K^−1^ and 367.98 W·m^−1^·K^−1^, respectively. Obviously, the thermal conductivity of 10% rGO/G-Cu is 21% higher than that of pure copper foil (342.47 W·m^−1^·K^−1^). Compared with the linear polymer polyvinyl alcohol, the two-dimensional planar polymer GO can directly act as a surfactant and “glue” not only to strengthen the bonding force between the thermal conductivity coating and the copper matrix but also to bond with each other through “hydrogen bond” force and interact with graphene through π–π conjugation to achieve uniform dispersion and stable distribution of graphene in the coating [47,50,51,52]. Concurrently, the GO in the coating is reduced to a certain extent under the catalysis of copper [35], thereby facilitating the repair of the interfacial region between the graphene sheet and the sheet. Ultimately, this results in a notable enhancement in the thermal conductivity of the cooling copper foil (rGO/G-Cu). More importantly, a comparative analysis was carried out with reference to the various copper-based thermal management materials (Table 1), and the composite film had a more environmentally friendly and efficient manufacturing method.

### 3.4. Practical Applications of Composite Films in LED Chips Heat Dissipation

To assess the thermal management ability of the prepared composite films for practical applications, an infrared thermal imager was employed to record the surface temperature of the composite materials during the operation of commercial LED chips (12 W). As illustrated in Figure 6a, Cu, 10% rGO/G-Cu, and 10% PVA/G-Cu composite films of the same size were assembled alongside the LED chips on identical devices (Figure 6b) separately. It is evident from Figure 6c that compared to copper foil, the real-time surface temperature of the composite films exhibited a higher rate of increase over time. The corresponding infrared thermal image depicting the operation process of the LED chips is presented in Figure 6d. Significantly, the surface temperature of 10% rGO/G-Cu reached 47.9 °C after 540 s operation, which is higher by 185.1% than the temperature observed with copper foil (16.8 °C). Furthermore, throughout the operational period of the LED chip, it can be observed that the real-time temperatures recorded for 10% rGO/G-Cu remained consistently higher than those for its counterpart 10% PVA/G-Cu with a maintained temperature gradient of approximately 2.3 °C. The high thermal conductivity of 10% rGO/G-Cu enabled efficient dissipation of heat generated during operation of the LED chip, indicating huge potential for the utilization of 10% rGO/G-Cu composite film for thermal management applications.

## 4. Conclusions

In summary, rGO/G-Cu films were successfully fabricated through a simple coating and rolling process by using GO as an adhesive and surfactant for graphene coating. The thermal conductivity test proves the excellent thermal conductivity of the composite material. Compared with pure Cu, the experimental results show that the thermal conductivity of 10% rGO/G-Cu film reached 414.56 W·m^−1^·K^−1^, which is 21% higher. Moreover, GO and graphene in 10% rGO/G-Cu showed a synergistic effect in drastically improving thermal conductivity compared to 10% PVA/G-Cu, where the conventional polymer PVA is used as the adhesive. The high thermal conductivity of 10% rGO/G-Cu is due to the incorporation of highly crystalline graphene with low defect content and excellent thermal conductivity in the film structure. Additionally, GO acts as a “patch” that is partially reduced under copper catalysis, thereby repairing defects and forming continuous channels for efficient heat transfer. Infrared imaging confirmed the excellent thermal conductivity performance of 10% rGO/G-Cu composite film, and the composite film also showed the ability to be fabricated on a large scale (Appendix A). Therefore, this environmentally friendly and cost-effective material has great potential for practical applications in thermal management systems. Furthermore, this study expands the concept of producing high-performance materials by employing graphene oxide as a dispersant and adhesive instead of traditional polymers.

## Figures and Tables

**Figure 1 polymers-16-01872-f001:**
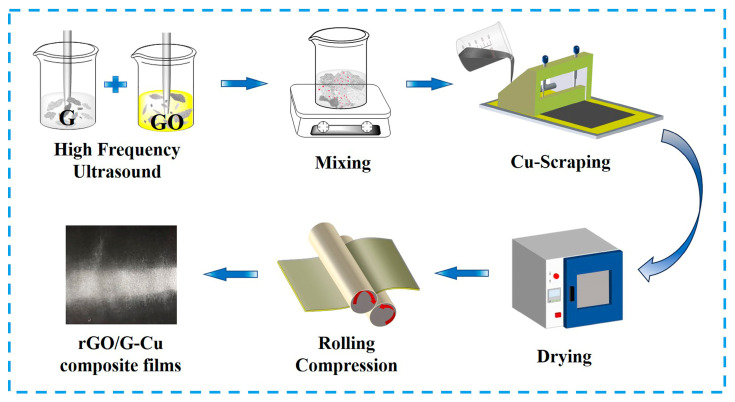
Fabrication scheme of rGO/G-Cu composite films.

**Figure 2 polymers-16-01872-f002:**
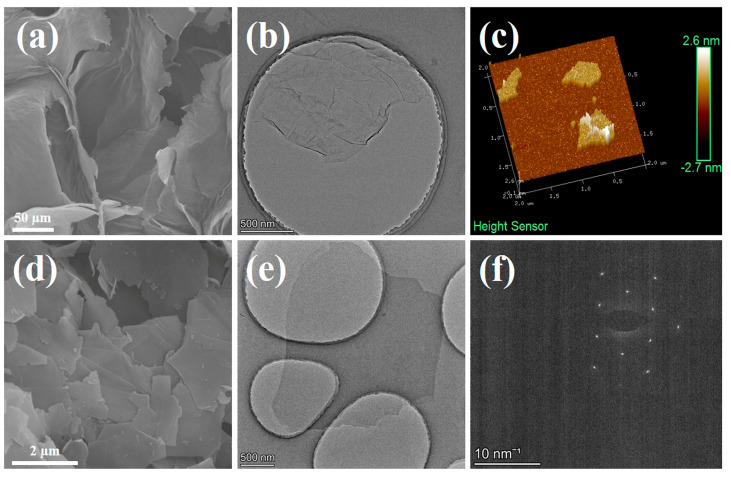
Scanning electron microscopy of GO (**a**) and graphene (**d**); TEM micrographs of GO (**b**) and graphene (**e**); AFM image showing thickness of GO (**c**); the electron diffraction pattern rings of near-single-layer graphene (**f**).

**Figure 3 polymers-16-01872-f003:**
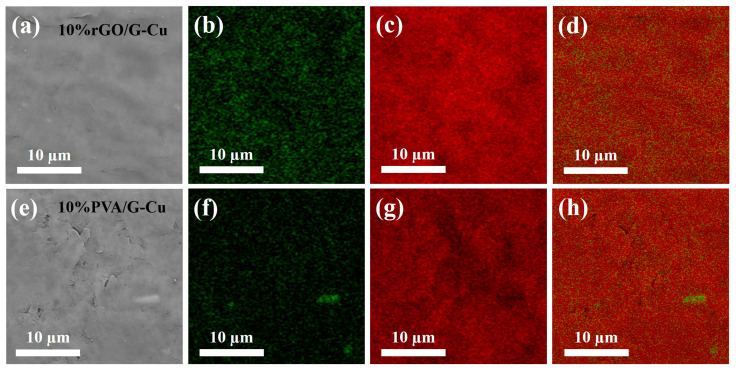
SEM image (**a**,**e**) and the corresponding EDS mapping (**b**–**d**), (**f**–**h**) of the composite films, O (green) and C (red).

**Figure 4 polymers-16-01872-f004:**
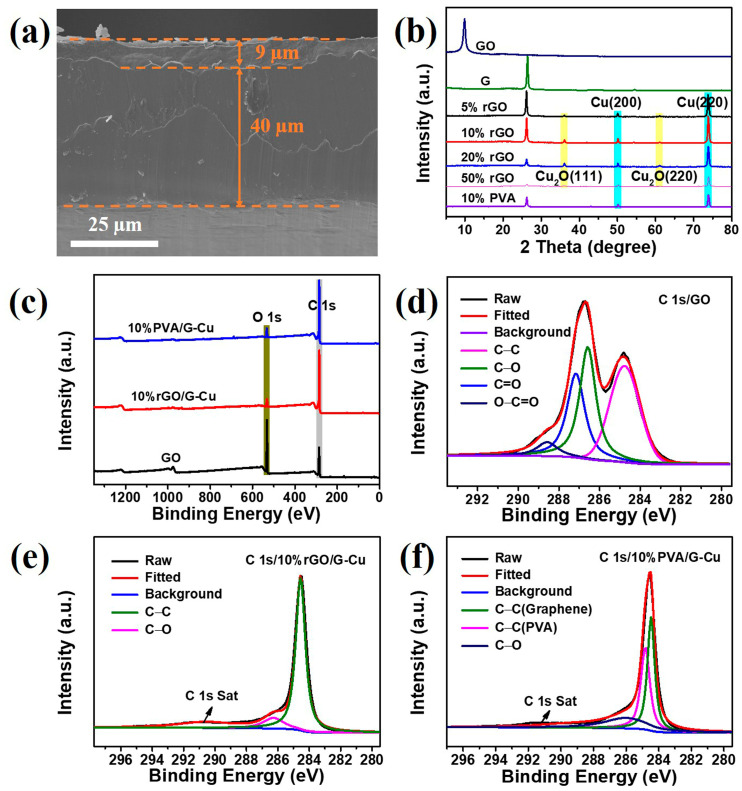
SEM images of the side (**a**) of 10% rGO/G-Cu; (**b**) XRD patterns of GO, graphene, and composite films; (**c**) XPS spectrum of the GO, 10% rGO/G-Cu, and 10% PVA/G-Cu; (**d**–**f**) high-resolution XPS spectra of C 1s.

**Figure 5 polymers-16-01872-f005:**
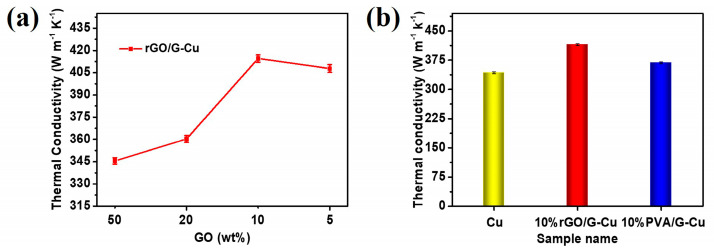
(**a**) Thermal conductivity of rGO/G-Cu composite film. (**b**) The thermal conductivity of the Cu foil, 10% rGO/G-Cu, and 10% PVA/G-Cu.

**Figure 6 polymers-16-01872-f006:**
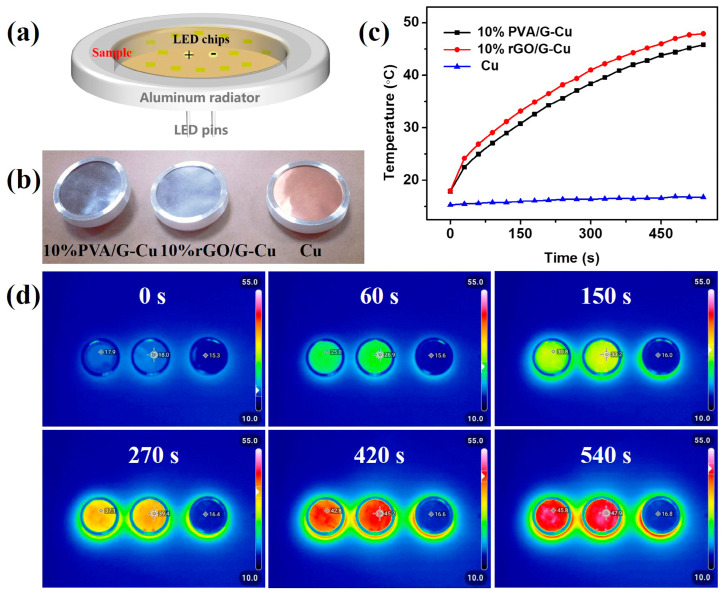
(**a**) Experimental setup of LED chips for thermal infrared imaging. (**b**) Optical images of Cu, 10% rGO/G-Cu, and 10% PVA/G-Cu. (**c**) The surface temperature of Cu, 10% rGO/G-Cu, and 10% PVA/G-Cu changes with the operating time of the LED chip from room temperature. (**d**) The corresponding infrared thermal images of the operation process of LED chips.

**Table 1 polymers-16-01872-t001:** Comparison of various copper-based thermal management materials.

NO.	Strengthening Phase	Additive	Method	K (W·m^−1^·K^−1^)	Reference
**1**	graphene nanosheets	carboxymethyl cellulose, conducting carbon black, and Styrene Butadiene rubber	coating	445.91	[15]
**2**	nitrogen-doped graphene nanosheets	carboxymethyl cellulose, conducting carbon black, and Styrene Butadiene rubber	coating	542.9	[15]
**3**	rGO	no	electrophoretic deposition and vacuum hot pressing	637.7	[20]
**4**	synthetic graphite	no	electroplating	526–626	[21]
**5**	graphene	no	vacuum filtration and spark plasma sintering	458	[34]
**6**	lignin-based nano-carbon	no	annealing treatment at 1000 °C in an argon atmosphere	478	[35]
**7**	graphene	no	chemical vapor deposition (1030 °C)	369.5	[36]
**8**	carbon nanofiber	no	electroless plating process and vacuum hot pressing	435	[53]
**9**	pure Cu foil	—	—	342.47	—
**10**	rGO/graphene	no	coating and rolling process	414.56	This work

## Data Availability

Data included in articles and Appendix A.

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
