# Peer review of "Enhancement of Thermal Management Performance of Copper Foil Using Additive–Free Graphene Coating"

_polymers, 2024, doi:10.3390/polym16131872_

Round 1
Reviewer 1 Report
Comments and Suggestions for Authors
Authors are thankworthy for their hard work to perform the research and writing the manuscript. I have some comments, suggestions and queries which are listed below:
1) Title should be revised, like “Enhancement of thermal management performance of copper foil using additive-free graphene coating.” Or, Thermal management performance of additive-free graphene-coated copper foil, or “effective approach for the preparation of graphene coated copper foil for obtaining better heat transfer,” etc.
2) Abstract should be revised focusing on the research background, methodology and findings of the work in brief. First few sentences are not needed in detailed.
3) Line 18: I couldn’t find the meaning of “graphene heat dissipation copper foil composite films (rGO/G-Cu)”. Please revise this sentence to make the articles easy to the readers. Moreover, what does, “G” with Cu means? If you mean “G” for graphene then you should mention it first and then use abbreviations (short forms)
4) Line 69: “The aforementioned present novel prospects for material design.” It seems there is a word missing after the word “aforementioned”. Please check it.
5) Line 96: ……washed with deionized water until natural pH value ‘achieved”.
6) Line 107: Authors added GO dispersion to graphene dispersion and stirred at room temperature for 24 hours (line 102 to 104). Then how rGO is obtained? Is only stirring enough to convert GO into rGO? No reducing agent is added to any step of GO/G mixture.
7) Line 107: rGO and G are very similar in terms of surface properties. In that case how authors have confirmed that graphene is attached to the Cu surface that is rGO/G-Cu. Why not G/rGO-Cu ?
8) How have authors added 10% rGO? I didn’t find any information related to the percent composition of rGO.
9) Line 109: There will be a full stop after “films”.
10) Line 109: “PVA/G-Cu composite films were prepared using the same method as a comparison”. Authors dispersed GO in aqueous medium at room temperature (line 100) but as I know PVA will not dissolve in water at room temperature. It requires heating at ~80 °C for melting. Then how both methods are the same?
11) Line 146-147: “As shown in Fig. 3 and Fig. S1, the composite films with varying GO contents exhibit uniformly flat and smooth surfaces, … “. How the authors vary GO contents, I didn’t find any information. There are two samples in Figure 3, one containing 10% rGO and another containing PVA. One plot is found in Figure 5 (a) which indicates 5, 10, 20, 50 % of GO. But there is no information given on how the percentage was calculated and how is the morphologies of other samples containing variable GO content. EDS data is supposed to reveal the elemental analysis like the percentage of C, O, H etc. but none is found over there.
12) There are huge discrepancies observe within the manuscript. Authors are requested to revise the manuscript carefully including all the necessary information.
Thanks.
Comments on the Quality of English Language
English language is good but there are scopes to improve the quality of English.
Author Response
A point-by-point reply to reviewers' comments:
Reply to Reviewer 1:
Thank you for your valuable feedback. We have carefully reviewed your comments and have rectified the corresponding shortcomings accordingly. The necessary corrections have been made in the revised version of the manuscript.
Q1) Title should be revised, like “Enhancement of thermal management performance of copper foil using additive-free graphene coating.” Or, Thermal management performance of additive-free graphene-coated copper foil, or “effective approach for the preparation of graphene coated copper foil for obtaining better heat transfer,” etc.
Reply: We have changed the title in the manuscript to “Enhancement of thermal management performance of copper foil using additive-free graphene coating”.
Q2) Abstract should be revised focusing on the research background, methodology and findings of the work in brief. First few sentences are not needed in detailed.
Reply: We have revised and optimized the abstract in manuscript.
Corrected contents in manuscript:
Advanced thermal interface materials with high thermal conductivity are crucial for addressing the heat dissipation issue in high-power, highly integrated electronic devices. One greatly potential way in this field is to take advantage of cooling copper foil (Cu) materials based on graphene(G). However, the current manufacturing of these cooling copper foil materials is accompanied by high cost, process complexity, and environmental problems,which limit their development and application. In this work, directly using graphene oxide as a dispersant and binder of graphene coating, a simplicity, low-cost, environmentally friendly graphene-copper foil composite films (rGO/G-Cu) with high thermal conductivity was successfully prepared. The microstructure characterization, thermal conductivity and thermal management performance tests were carried out for the composite films. These results demonstrate that compared to pure copper foil (342.47 W·m-1·K-1) and 10% PVA/G-Cu (367.98 W·m-1·K-1) with polyvinyl alcohol as a binder, 10% rGO/G-Cu exhibits better thermal conductivity (414.56 W·m-1·K-1). The introduction of two-dimensional graphene oxide effectively enhance the adhesion between the coating and the copper foil, while greatly improving its thermal conductivity. Furthermore, experimental results indicate that rGO/G-Cu exhibited excellent heat transfer performance and flexibility. This work is highly meaningful for developing economical and environmentally friendly materials with high thermal conductivity to meet the increasing demand for heat dissipation.
Q3) Line 18: I couldn’t find the meaning of “graphene heat dissipation copper foil composite films (rGO/G-Cu)”. Please revise this sentence to make the articles easy to the readers. Moreover, what does, “G” with Cu means? If you mean “G” for graphene then you should mention it first and then use abbreviations (short forms)
Reply: We have revised this sentence in our revised manuscript (line 14 ~ 15). "G" and "Cu" stand for graphene and copper foil, respectively. We have mentioned them in line 10 ~ 11 and then use abbreviations.
Q4) Line 69: “The aforementioned present novel prospects for material design.” It seems there is a word missing after the word “aforementioned”. Please check it.
Reply: I'm sorry. It's our oversight. We have checked it in our revised manuscript. (line 65)
Q5) Line 96: ……washed with deionized water until natural pH value “achieved”.
Reply: We have replaced “until” with “achieved” in our revised manuscript. (line 93)
Q6) Line 107: Authors added GO dispersion to graphene dispersion and stirred at room temperature for 24 hours (line 102 to 104). Then how rGO is obtained? Is only stirring enough to convert GO into rGO? No reducing agent is added to any step of GO/G mixture.
Reply: Studies have indicated that the presence of a Cu substrate facilitates the reduction of graphene oxide. During vacuum drying at 95℃, GO was reduced to rGO in situ due to the catalysis of Cu substrate. We have explained the origin of rGO in part of Preparation of rGO/G-Cu Composite Films (line 103 ~ 105).
Q7) Line 107: rGO and G are very similar in terms of surface properties. In that case how authors have confirmed that graphene is attached to the Cu surface that is rGO/G-Cu. Why not G/rGO-Cu ?
Reply: As you mentioned, graphene and reduced graphene oxide (rGO) are both two-dimensional lamellar structural materials with similar surface properties. The coating on the copper foil surface is formed by graphene and reduced graphene oxide sheets on top of each other, so rGO/G and G/rGO are both correct expressions, and we chosed one of them.
Q8) How have authors added 10% rGO? I didn’t find any information related to the percent composition of rGO.
Reply: We have added information of the percent composition of rGO. Since the mass of graphene oxide changes little after reduction to reduced graphene oxide, the mass of graphene oxide is taken as statistics. The mass fraction (wt%) of GO (rGO) was calculated by the following equation:
Here MGO, MG, and MrGO represent the mass of GO, graphene (G), and rGO, respectively. Then the mass fraction of rGO in the composites was 5, 10, 20, and 50 wt%, respectively. (line 106 ~ 110)
Q9) Line 109: There will be a full stop after “films”.
Reply: We have revised it in our revised manuscript. (line 111)
Q10) Line 109: “PVA/G-Cu composite films were prepared using the same method as a comparison”. Authors dispersed GO in aqueous medium at room temperature (line 100) but as I know PVA will not dissolve in water at room temperature. It requires heating at ~80 °C for melting. Then how both methods are the same?
Reply: We have added PVA solution preparation part. Polyvinyl alcohol (PVA) needs to be pre-dissolved in 85 â—¦C deionized water, then cooled to room temperature. (line 112 ~ 114)
Q11) Line 146-147: “As shown in Fig. 3 and Fig. S1, the composite films with varying GO contents exhibit uniformly flat and smooth surfaces, … “. How the authors vary GO contents, I didn’t find any information. There are two samples in Figure 3, one containing 10% rGO and another containing PVA. One plot is found in Figure 5 (a) which indicates 5, 10, 20, 50 % of GO. But there is no information given on how the percentage was calculated and how is the morphologies of other samples containing variable GO content. EDS data is supposed to reveal the elemental analysis like the percentage of C, O, H etc. but none is found over there.
Reply: No information was given on how the percentages were calculated and what the morphology of other samples containing variable GO conten, which was our oversight. We have explained the calculation method of graphene oxide or rGO content in Q 8, and added the surface morphology and element distribution of 50% rGO/GO-Cu in Fig. S1. EDS elemental analysis of all samples is also supplemented in Table S1, and the analysis results are supplemented. (line 170)
Q12) There are huge discrepancies observe within the manuscript. Authors are requested to revise the manuscript carefully including all the necessary information.
Reply: We accordingly checked our manuscript carefully and made the corresponding modifications.

Reviewer 2 Report
Comments and Suggestions for Authors
The goal of the presented work was to develop a coating with high thermal conductivity, ensuring the maintenance of an acceptable thermal balance on devices such as powerful processors. The specific task of the work was to develop a pure graphene coating on copper foil, for which the authors used a composition of two 2D graphene structures: graphene itself (G) and graphene oxide (GO). GO acts as a binder for G sheets and improves adhesion to the copper substrate. Changing the GO content in the composition allowed the authors to find the optimal formulation of the mixture with maximum thermal conductivity. A comparison of the data obtained by the authors with the literature suggests that the approach proposed by the authors to the creation of thermally conductive coatings is promising.
The article meets the profile of the Journal and can be recommended for publication after correcting a number of shortcomings in the manuscript:
1. The authors should provide XRD and Raman spectroscopy data for both graphene obtained from an external source and GO synthesized by the authors. From this data, they must calculate the number of layers in the resulting 2D graphene structures and their defectiveness.
2. There is no brand and characteristics of the polyvinyl alcohol used in the work.
3. The manuscript does not provide the parameters of the ultrasonic device.
Author Response
A point-by-point reply to reviewers' comments:
Reply to Reviewer 2:
Thank you for your valuable feedback. We have carefully reviewed your comments and have rectified the corresponding shortcomings accordingly. The necessary corrections have been made in the revised version of the manuscript.
Q1. The authors should provide XRD and Raman spectroscopy data for both graphene obtained from an external source and GO synthesized by the authors. From this data, they must calculate the number of layers in the resulting 2D graphene structures and their defectiveness.
Reply: We have provided XRD spectroscopy data of graphene obtained from an external source and GO synthesized in Fig. 4f, and Raman spectroscopy data to determine the degree of defect of graphene and graphene oxide were added to Figure S1(m, n). We have revised the manuscript of the extent of defects in graphene and graphene oxide. (Line 137-142).
Corrected contents in manuscript:
As shown in Fig. S1(m, n), the weak D peak represents sp3 carbon in the defect domain, and the sharp G peak represents sp2 carbon in the graphite lattice domain, the intensity ratio of peak D to peak G (ID/IG) is used to evaluate the defect grade of carbon materials. The ID/IG value of GO is as high as 0.85, indicating a large number of defects due to oxidation. In contrast, the graphene has an extremely low ID/IG value of 0.06, indicating high-quality graphene with fewer defects.
For the calculation of the number of layers of the two-dimensional graphene structure of the graphene powder that has not been further stripped, we believe that if the calculation is performed using XRD and Raman spectroscopy data, it will get a very fuzzy result, so we use TEM and AFM, two more intuitive characterization methods to determine the number of layers of graphene. (Line 144-148).
Q2. There is no brand and characteristics of the polyvinyl alcohol used in the work.
Reply: We have mentioned in the materials section the brand and characteristics of the polyvinyl alcohol used in this work, and added the full name of polyvinyl alcohol. (line 81 ~ 83)
Q3. The manuscript does not provide the parameters of the ultrasonic device.
Reply: We have added the parameters of the ultrasonic device in our revised manuscript. (line 96)

Round 2
Reviewer 1 Report
Comments and Suggestions for Authors
Thank authors for their smart reply to the reviewer’s comments/queries and revision of their manuscript. The quality of the manuscript has been improved though there are some observations remaining which are listed below:
Line 13-15:
The highlighted sentence can be revised as:
In this work, a simple, low-cost, environmentally friendly graphene-copper foil composite films (rGO/G-Cu) with high thermal conductivity was successfully prepared using graphene oxide directly as a dispersant and binder of graphene coating.
Line 92-93:
“The obtained sample was centrifuged and washed with deionized water achieved natural pH value” should be “The obtained sample was centrifuged and washed with deionized water to achieve natural pH value.
Line 96-97:
“GO was dispersed in deionized water (50 mL) by high-frequency ultrasound for 10 min at room temperature, resulting in a clear GO solution.” How much GO was dispersed? As obtained GO after centrifuge (line 92-93)? Is so then the amount of GO would be <4 g (amount of graphite-Line 86)
Line 103-105:
The revised additional information requires a citation of reference supporting that Cu can cause reduction of GO on drying at 95°C.
Line 103-110:
The equation is confusing because this equation is used to find out the fraction of components in a mixture in wt. %. Equation 1 indicates the percentage of GO in a mixture of GO and G (middle) and in a mixture of rGO and G (right) and these two parts are the same. Besides, that is the wt. % of Graphene (G) and what is represented by x in equation 1?
Line: 170:
Table S1 shows that the total wt. % of C and O element is 100. But as I know, there must be some hydrogen remaining in rGO after reducing GO. Because GO contains -OH (Hydroxyl group), -COOH (carboxyl group), -O- (epoxide group) etc. So, there must be a small amount of hydrogen which is missing. Interestingly, the sample containing 10wt.% PVA has a total of 100% C and O elements, but PVA contains hydrogen. Where is the wt. % of hydrogen? You need not show wt.% of H, but total of C and O must be less than 100% keeping wt. % of H aside.
Figure S1: O (green) and C (red) but what are (d), (h), and (i)?
Comments on the Quality of English LanguageEnglish seems good but there are scopes to improve the quality of English Language. I have suggested the corrections of some sentences. Authors should go through the manuscript and revise English wherever needed.
Author Response
A point-by-point reply to reviewers' comments:
Reply to Reviewer 1:
Thank you for your valuable feedback. We have carefully reviewed your comments and corrected the corresponding deficiencies accordingly. We have read through the entire manuscript and made the necessary changes to improve the quality of the English language. In this revised version, changes to manuscript within the document were all highlighted by using red colored text.
- Line 13-15:The highlighted sentence can be revised as: In this work, a simple, low-cost, environmentally friendly graphene-copper foil composite films (rGO/G-Cu) with high thermal conductivity was successfully prepared using graphene oxide directly as a dispersant and binder of graphene coating.
Reply: Thank you for reviewer’s good suggestion. We have changed the highlighted sentence in the manuscript to “In this work, a simple, low-cost, environmentally friendly graphene-copper foil composite films (rGO/G-Cu) with high thermal conductivity was successfully prepared using graphene oxide directly as a dispersant and binder of graphene coating.” (Line 13-15)
- Line 92-93: “The obtained sample was centrifuged and washed with deionized water achieved natural pH value” should be “The obtained sample was centrifuged and washed with deionized water to achieve natural pH value.
Reply: Thank you for reviewer’s good suggestion. We have changed the sentence in the manuscript to “The obtained sample was centrifuged and washed with deionized water to achieve natural pH value. ” (Line 92-93)
- Line 96-97: “GO was dispersed in deionized water (50 mL) by high-frequency ultrasound for 10 min at room temperature, resulting in a clear GO solution.” How much GO was dispersed? As obtained GO after centrifuge (line 92-93)? Is so then the amount of GO would be <4 g (amount of graphite-Line 86)
Reply: Thank you for your question. The process of preparing 10% rGO/G-Cu is taken as a typical operation flow, 100 mg GO was dispersed in deionized water (50 mL) by high-frequency ultrasound, and we have added this important information in our manuscript. (Line 97)
In addition, the GO used for the preparation of rGO/G-Cu is derived from graphene oxide after centrifugation.
- Line 103-105: The revised additional information requires a citation of reference supporting that Cu can cause reduction of GO on drying at 95°C.
Reply: Thank you for your reminder. The related reference has been added in the manuscript. (Line 105 in manuscript)
Voylov, D.N.; Agapov, A.L.; Sokolov, A.P.; Shulga, Y.M.; Arbuzov, A.A. Room temperature reduction of multilayer graphene oxide film on a copper substrate: penetration and participation of copper phase in redox reactions. Carbon 2014, 69, 563–570.
- 5. Line 103-110: The equation is confusing because this equation is used to find out the fraction of components in a mixture in wt. %. Equation 1 indicates the percentage of GO in a mixture of GO and G (middle) and in a mixture of rGO and G (right) and these two parts are the same. Besides, that is the wt. % of Graphene (G) and what is represented by x in equation 1?
Reply: Thank you for your reminder. We apologize for the confusion caused by our vague definition of x. x represents the mass fraction of GO (rGO) in the graphene coating, which we have modified in manuscript. (Line 111)
- 6. Line: 170: Table S1 shows that the total wt. % of C and O element is 100. But as I know, there must be some hydrogen remaining in rGO after reducing GO. Because GO contains -OH (Hydroxyl group), -COOH (carboxyl group), -O- (epoxide group) etc. So, there must be a small amount of hydrogen which is missing. Interestingly, the sample containing 10wt.% PVA has a total of 100% C and O elements, but PVA contains hydrogen. Where is the wt. % of hydrogen? You need not show wt.% of H, but total of C and O must be less than 100% keeping wt. % of H aside.
Reply: It is true that H is definitely contained in the composite film. Since H does not have inner electrons, it is not used for quantitative analysis in EDS elemental analysis. We only conduct quantitative analysis of C and O elements in the composite film, so the sum of the mass fraction of C and O is 100%.
- 7. Figure S1: O (green) and C (red) but what are (d), (h), and (i)?
Reply: Thanks for your good question. Figure S1(d), (h) and (i) show the distribution of elements C and O in EDS mapping of composite films.
